# Unveiling the Genetic Symphony: Harnessing CRISPR-Cas Genome Editing for Effective Insect Pest Management

**DOI:** 10.3390/plants12233961

**Published:** 2023-11-24

**Authors:** J. Komal, H. R. Desai, Ipsita Samal, Andrea Mastinu, R. D. Patel, P. V. Dinesh Kumar, Prasanta Kumar Majhi, Deepak Kumar Mahanta, Tanmaya Kumar Bhoi

**Affiliations:** 1Basic Seed Multiplication and Training Centre, Central Silk Board, Kharaswan 833216, Jharkhand, India; komalchary3498@gmail.com; 2Department of Entomology, Main Cotton Research Station, Navsari Agricultural University, Surat 395007, Gujarat, India; hrdesai@nau.in (H.R.D.); rdpatel@nau.in (R.D.P.); 3Indian Council of Agricultural Research-National Research Centre on Litchi, Mushahari, Ramna, Muzaffarpur 842002, Bihar, India; happyipsu29@gmail.com; 4Department of Molecular and Translational Medicine, Division of Pharmacology, University of Brescia, 25123 Brescia, Italy; 5Research Extension Centre, Central Silk Board, Hoshangabad 461001, Madhya Pradesh, India; dineshgupta1539@gmail.com; 6Department of Plant Breeding and Genetics, Odisha University of Agriculture and Technology, Bhubaneswar 751003, Odisha, India; prasantakumarmajhi53@gmail.com; 7Forest Entomology Discipline, Forest Protection Division, Indian Council of Forestry Research and Education (ICFRE)-Forest Research Institute (ICFRE-FRI), Dehradun 248006, Uttarakhand, India; 8Forest Protection Division, Indian Council of Forestry Research and Education (ICFRE)-Arid Forest Research Institute (ICFRE-AFRI), Jodhpur 342005, Rajasthan, India

**Keywords:** genome editing, CRISPR-Cas, insects, crops, management, stress

## Abstract

Phytophagous insects pose a significant threat to global crop yield and food security. The need for increased agricultural output while reducing dependence on harmful synthetic insecticides necessitates the implementation of innovative methods. The utilization of CRISPR-Cas (Clustered regularly interspaced short palindromic repeats) technology to develop insect pest-resistant plants is believed to be a highly effective approach in reducing production expenses and enhancing the profitability of farms. Insect genome research provides vital insights into gene functions, allowing for a better knowledge of insect biology, adaptability, and the development of targeted pest management and disease prevention measures. The CRISPR-Cas gene editing technique has the capability to modify the DNA of insects, either to trigger a gene drive or to overcome their resistance to specific insecticides. The advancements in CRISPR technology and its various applications have shown potential in developing insect-resistant varieties of plants and other strategies for effective pest management through a sustainable approach. This could have significant consequences for ensuring food security. This approach involves using genome editing to create modified insects or crop plants. The article critically analyzed and discussed the potential and challenges associated with exploring and utilizing CRISPR-Cas technology for reducing insect pest pressure in crop plants.

## 1. Introduction

The global agricultural yield undergoes a substantial annual decline of around one-fifth (20%) because of the adverse effects caused by phytophagous insects [1,2]. These insects feed on and damage crops, resulting in significant economic losses and food insecurity [3,4,5]. Any country’s ability to produce enough food is hampered by biotic stress [6,7] since it reduces both the quality and quantity of crops [8]. According to [9], the yield loss brought on by insect infestation has terrible effects on society, including starvation and poverty. Insect pests are reemerging and becoming invasive, and this, coupled with a fast-expanding human population, needs urgent innovations as well as strict and coordinated agricultural methods. According to the FAO’s estimation [10], plant diseases and insect pests reduce agricultural output globally by 20–40% annually. In the era of climate change and uncertain weather patterns, researchers anticipated a sharp reduction in agricultural output [11,12]. According to [13], climatic changes may make phytophagous pests more dangerous to crops by raising the risk they pose. Chemical control is a rapid and effective way to get rid of diseases, but its negative effects on the environment and the development of resistance raise severe questions that restrict its usage [14]. Biological control is used as a substitute since it is safe for the environment. Due to its poor effectiveness, inconsistent application, and high cost, its usage is currently constrained [6]. Conventional breeding has significantly contributed to global food security over the last several decades [15]. However, this method has a few drawbacks. The first is the species barrier, which prevents the introduction of novel features to a particular species of interest since two plants from different species cannot often cross it [16]. The second is the development of undesired consequences because linkage drag causes other characteristics to transmit along with the intended one [16]. By utilizing physical or chemical mutagens (5 Bromo Uracil, ethyl methane sulphonate, and methyl methane sulphonate), plants with induced genetic variations were produced through mutation breeding [17]. However, mutagenesis is a random process that results in several undesirable genetic changes. One of the most advanced and preferred techniques for enhancing crops with desired traits is transgenic development by inserting functional genetic components [18]. Transgenesis has several benefits, one of which is the ability to cross the species barrier and make use of the entire gene pool. But despite being subject to strict regulatory laws, transgenic crops, often referred to as GM (genetically modified) crops, remain divisive due to a lack of public knowledge and inaccurate portrayals by a number of NGOs and anti-GM activists [6].

In recent years, there has been a notable increase in the focus on insect genome studies and the comprehension of gene functions. Numerous initiatives, such as the i5k initiative, have played a vital role in enhancing our knowledge within this domain. The i5k initiative, also known as the “5000 Insect Genome Project,” was initiated with the objective of conducting genome sequencing and analysis on a total of 5000 insect species [19]. The principal objective of the project is to facilitate the extensive examination of insect genomics and offer valuable resources for comprehending insect biology, evolution, and ecology [20,21]. Through the examination of the genomes of various insect species, researchers are able to discern shared genetic characteristics, distinctive adaptations, and evolutionary trends within disparate lineages of insects. Genome editing (GE) has gained significant popularity as a novel breeding technology (NBT). As per [22], it can be observed that, unlike genetically modified crops, a majority of the genome editing-based cultivars do not possess any exogenous DNA and are not subjected to supplementary regulatory measures in several nations. Genome editing, commonly known as gene editing, involves the insertion, deletion, labeling, or reordering of genetic material to produce a desired trait through genetic manipulation [23]. To date, the primary sequence-specific nucleases utilized in gene editing can be grouped into four main categories, namely mega nucleases (MNs), transcription activator-like effector nucleases (TALENs), zinc finger nucleases (ZFNs), and clustered regularly interspaced short palindromic repeats (CRISPR)/CRISPR-associated proteins (Cas) [24]. The latest and most advanced genome-editing technology is the CRISPR-Cas RNA-guided nucleases, which have been adapted from bacterial innate immune mechanisms and are based on Type II CRISPR-Cas9 mechanisms [25]. According to [26], the superiority of CRISPR-Cas technology over MNs, ZFNs, and TALENs can be attributed to its user-friendliness, adaptability, and capability for multiple gene insertion and editing in a single run. Modern crop improvement methods, including genetically modified crops and stress-resistant crops, are prioritized in order to ensure food supply for the world’s rising population and to achieve Sustainable Development Goal 2 (zero hunger) [27]. It is costly to employ synthetic chemical pesticides to manage insect pests during agricultural production and dangerous for both people and the environment [28]. Furthermore, it negatively causes off-target effects and poses a threat to biodiversity. Numerous insect resistance genes, such as *Bacillus thuringiensis* (*Bt*)-insecticidal crystal proteins (ICPs), have previously been used in genetically modified crops, which has had a substantial impact on production and sustainability. CRISPR-Cas gene editing is a viable technique for producing insect-resistant plants that will promote sustainable agriculture [27]. By changing effect or target interactions, removing host-susceptible genes, decoupling the detrimental impact of defense hormones, and other methods, it may be possible to develop insect resistance using this prospective technology [29]. Insect-resistant plants and various insects have been modified using CRISPR-Cas gene editing over the past ten years. Using this strategy, sustainable control of insect pests has shown significant potential for boosting agricultural productivity. The production of agricultural plants and insects with altered genomes may help with insect pest management in agriculture. The goal of this study was to provide a brief overview of the most recent advances in the use of CRISPR-Cas to modify the genomes of phytophagous insects and produce plants that are resistant to insects. We also spoke about the challenges and prospective applications of the CRISPR-Cas toolkit for the long-term control of insect pests.

## 2. Understanding the Mechanism of CRISPR-Cas System

CRISPR refers to clustered, regularly interspaced short palindromic repeats, while Cas denotes the associated protein. Prokaryotic organisms, including bacteria (45%) and archaea (84%), possess an intrinsic defense mechanism encoded in their genomes [30]. CRISPR-Cas9 is a gene-editing technique that enables the precise cutting of a specific DNA section, resulting in the inactivation of the gene or the replacement of one gene version with another (Figure 1) [31]. In 2012, Emmanuelle Charpentier, Jennifer Doudna, and Fang Zhang invented it, and in 2020, the former two scientists received the Nobel Prize (Chemistry) [32]. This represents the latest gene editing technology. CRISPR-Cas9 technology is classified into three types: I, II, and III. In type II systems, the Cas9 nuclease necessitates DNA that corresponds to a single RNA guide (sgRNA) [33]. The CRISPR-Cas system comprises the guide RNA and the Cas protein as its main constituents. The Cas9 protein, known as a nuclease enzyme or molecular scissors, cleaves DNA. Guide-RNAs facilitate the targeted delivery of Cas9 to a specific site in the genome, enabling the replacement of the existing sequence with a new one [34]. Molla et al., 2020 [35] reported that it is a highly efficient and rapid genome-editing tool. The CRISPR-Cas technology enables multiplexing or epigenetic gene editing (where trait function can be reversible, a chance to originality possible).

Zuo et al., 2020 [36] has demonstrated various CRISPR-Cas applications that modify insect or plant genomes’ DNA sequences. *Streptococcus pyrogene* (*Sp*) is the primary origin of the Cas9 protein, which is widely utilized [37]. The process involves the use of a Cas9-protein associated single-guide RNA (sgRNA) to cleave a specific target DNA region adjacent to a protospacer adjacent motif (PAM), resulting in the activation of the cellular DNA repair system to generate a Double-Strand Breaks (DSB). In the absence of non-homologous end-joining (NHEJ), Homology-Directed Repair (HDR) pathways are activated, leading to spontaneous insertions, replacements, or deletions at the site of double-strand breaks (DSBs). This often results in the disruption of gene function. Alternatively, HDR mechanisms can be activated to achieve precise gene alterations such as knock-outs, knock-ins, or mutations, provided that donor DNA templates are available and similar to the sequence surrounding the DSB site [38]. The HDR and NHEJ mechanisms have been utilized for genome editing in insects and plants [39]. The CRISPR-Cas construct is introduced into target cells using particle-bombardment-mediated transformation methods or Agrobacterium-mediated and into insect embryos using transfusion, microinjection, or electroporation-mediated transformation methods to generate transgenic species with desired traits [40].

## 3. Potentiality of Revolutionizing Pest Management Using CRISPR-Cas Genome Editing

The utilization of biotechnology is of paramount importance in managing insect pests to safeguard crops and enhance yields. This encompasses a broad spectrum of activities, ranging from the development of pest-resistant breeds to the genetic modification of novel genes through introgression [41]. The developing stage of utilizing genome-editing methodologies for insect-resistant plants is currently underway. Genome editing has the potential to regulate insect populations through the manipulation of genes in both plants and insects (Figure 2), as noted by [42].

According to [43], when there are insufficient R-genes (Resistant genes), de novo resistance can be produced in order to control insect pests in crops. Other methods include causing insect pest sterility or disrupting the resistance gained against pesticides. The utilization of CRISPR-Cas9 genome editing technology is currently being investigated to alter insects in order to inhibit their ability to feed on and harm plants, as well as to alter and modify plants to enhance their effectiveness in deterring insects, as per the research conducted by [44]. A promising method to develop customized plants is the genome-editing platform, especially when a particular deletion can enhance desired features or trigger a gene drive to spread mutations that reduce the survival of female insects [45]. The issue of insect resistance to GM crops having *Bt* endotoxins poses a threat to the agricultural biotechnology industry. As a result, biotech companies are seeking an innovative and alternative solution that is both economically feasible and environmentally sustainable. CRISPR-Cas9 gene editing is the primary technique utilized in the biotech industry for managing insect pests [46]. Genome-editing technology utilizes the cell’s internal processes to modify a gene’s function. Genome editing involves modifying a target genome’s DNA sequence by including new DNA bases or removing or replacing existing ones [47].

### 3.1. Precision CRISPR/Cas-Mediated Genome Editing at the Insect Level

Since the CRISPR/Cas9 gene-editing approach was first used in mammalian cells in 2012, the fields of molecular biology and biotechnology have made phenomenal advancements and transformations [48]. The CRISPR/Cas9 system has gained significant attention as a promising genome editing tool following the identification of the homology-dependent cleavage recombination mechanism. The increasing application of gene-editing tools as a precise method for editing genomes presents numerous possibilities that could potentially affect significant agronomic characteristics, including resistance to biotic stresses, as evidenced in studies conducted by [49,50]. This study provides an overview of the potential applications of the CRISPR/Cas9 system and its recent prominence in the field of insect pest management. The system’s unique advantages have garnered significant attention due to its potential to enhance crop yield and ensure food security. Gene editing techniques have been employed in various insect species, including *Drosophila melanogaster* Meigen, several fruit flies and mosquitoes, to address fundamental inquiries regarding insect biology. In contemporary times, technology has been employed to create innovative methods for controlling pests [51], and it has demonstrated its effectiveness in managing pests [52]. The progressions in genome editing techniques have facilitated the emergence of innovative pest management strategies through the development of insects that are genetically modified. The CRISPR/Cas technology is currently undergoing advancements due to its advantageous capabilities in manipulating genes precisely. As per [53], the CRISPR/Cas tool’s constituents, namely the Cas9 protein, and sgRNA, can be introduced into the intended organism in the form of a ribonucleo-protein (RNP) complex, plasmid DNA, or RNA.

#### 3.1.1. CRISPR/Cas-Mediated Genome Editing in Lepidopteran Insects

Lepidoptera has a variety of taxa and is one of the most well-known insect order globally [54]. Lepidopterans have both scientific and commercial significance since they are the second most invasive pests of agricultural items that have been preserved [55]. Despite the high level of interest in this group, there has been insufficient advancement in the area of genetic modification. Crop yields are drastically impacted by lepidopterans.

The principal Lepidopteran insect pests, such as *Helicoverpa armigera*, *Plutella xylostella, Spodoptera frugiperda,* and *Spodoptera litura*, were too controlled with chemicals, which caused the insects to become resistant to conventional pesticides [56]. Resistance to gene silencing (RNAi) and susceptibility to inbreeding were the two main factors that led to the failure of the systematic use of biotechnological tools in Lepidoptera [57]. Fortunately, CRISPR/Cas9-mediated GE offered a heritable and environmentally acceptable approach to pest control in lepidopterans (Figure 3). The GE methods employed to study gene functions in some of the lepidopteron insects are well analyzed by [58]. According to [59], insects rely on dietary sources to fulfill their sterol requirements due to their inability to synthesize them endogenously. The authors in [60] have identified *NPC1b* as an insect protein that functions as an integral membrane protein. Moreover, the authors in [59] have reported that *NPC1b* plays a crucial function in the absorption of dietary cholesterol in *Drosophila melanogaster*. The study utilized CRISPR/Cas9-edited *NPC1b* mutant larvae to investigate the role of *NPC1b* in *Helicoverpa armigera*. The significance of *NPC1b* in the growth and uptake of cholesterol through diet has been established in *H. armigera*. It has been observed that a restriction in the dietary uptake of cholesterol leads to a hindrance in the insect’s weight gain and food ingestion, as reported by [60]. Therefore, *NPC1b* has the potential to serve as a target for pest management. Nevertheless, it is possible that the method could result in unintended consequences [61]. The utilization of the CRISPR/Cas9 system in *H. armigera* by [62] demonstrated that *HaCad*, a crucial receptor of Cry1Ac, is responsible for developing resistance to Cry1Ac.

The approach employed involved the injection of a combination of Cas9 mRNA and sgRNA into the eggs, as opposed to the introduction of plasmids encoding Cas9 and sgRNA. The outcome of this method was a remarkably effective modification of the *HaCad* locus. This investigation facilitated a comprehensive understanding of the development of resistance to endotoxins in *Helicoverpa* and alternative ways to overcome it as reported by [62]. The utilization of the CRISPR/Cas9 system has presented a novel strategy for pest management in *H. armigera* through the optimization of mating time via antagonist-mediated means, resulting in the attainment of minimal fecundity levels [63]. In both *H. armigera* and *H. punctigera*, the resistance developed to the Bt toxin Cry2Ab has been associated with a mutation resulting in the loss of function of a transporter gene, ABC (*ABCA2*). HaABCA2 knock-out strains were generated using the CRISPR/Cas9 technique in order to establish a link between Cry2Ab resistance in *H. armigera* and the ABCA2 gene. According to [64], the knock-out strain exhibited a considerable degree of resistance, thereby validating the involvement of *HaABCA2* in mediating the toxicity of Cry2Ab and Cry2Aa against *H. armigera*. The authors in [65] utilized the CRISPR/Cas9 system to establish mutants bearing four pigment genes, namely scarlet, ok, white, and brown in *H. armigera*. This resulted in various phenotypic alterations with physiological implications. To enhance comprehension of the reaction between insects and insecticides, as well as the control of agricultural pests, a group of nine P450 genes in *H. armigera* were subjected to CRISPR/Cas9-mediated knock-out lines. The research conducted by [66] furnished fundamental data for the meticulous assessment and recognition of the principal factors involved in insecticide metabolism.

The *Slabd-A* gene (*S. litura* abdominal-A) was targeted using CRISPR/Cas9-mediated mutagenesis on Lepidopteran pests. The *Slabd-A* gene plays a crucial role in determining segment identity and abdominal segmentation in insects. *Slabd-A* plays a role in its embryonic development stage. Genome manipulation targeting this gene led to abnormal segmentation and pigmentation, as reported by [67]. Zhu et al., 2017 [68] found that knocking out the *SlitBLOS2* gene in *S. litura* led to the disappearance of significant markings on the integument, including the yellow strips and white spots in the larval stage. This gene can serve as an important marker for functional studies and pest control strategies [69]. Through the study of the genetic component, researchers can acquire a deeper understanding of the intricate developmental procedures and molecular mechanisms that underlie these specific physical traits. If the *SlitBLOS2* gene or its associated pathways are essential for the development of these traits, it could be feasible to selectively target them as a potential strategy for managing populations of *S. litura*. [9] identified PBPs (Pheromone Binding Proteins) as the primary odor-binding proteins. The CRISPR/Cas9 system was utilized to mutate the *SlitPBP3* gene in *S. litura* for the purpose of illustrating its function. The authors in [70] observed a significant reduction in attraction to sex pheromone components in *SlitPBP3* mutants compared to wild-type individuals. The CRISPR/Cas9 system was utilized to obtain knock-outs of the olfactory receptor coreceptor (*Orco*) gene in *Spodoptera littoralis*. The knock-outs exhibited an inability to react to sex pheromones as well as plant odors, as reported by [71]. A study focused on three genes in *S. frugiperda*. The study focused on three genes: tryptophan 2, 3-dioxygenase (TO) and biogenesis of lysosome-related organelles complex 1 subunit 2 (*BLOS2*), which are marker genes, and *E93*, a developmental gene that promotes growth in adult insects by inducing ecdysone. This study proposed a technique involving multiple sgRNA injections. The findings may have practical applications in functional gene characterization and high-throughput functional genomics screening, which could enhance our comprehension of the heritable mechanisms governing the critical pathways of inducing anosmia in adult moths of Fall Army Worm (FAW) and other invasive pests. The authors in [36] utilized the CRISPR/Cas9 system to functionally validate *Seα6-KO* (Homozygous strain) in *S. exigua*.

The insect’s extracellular matrix (ECM) comprises chitin and protein. Chitin is a bio-polymer produced by the *CHS1* gene-encoded protein. Benzoylureas (BPUs), etoxazole, and buprofezin are chemicals that inhibit chitin synthesis. The authors in [72] reported a single nucleotide polymorphism (SNP) at position 1017 of the CHS1 gene, resulting in the swapping of isoleucine (I) with phenylalanine (F). This substitution was found to be associated with etoxazole resistance, which led to a better understanding of the mode of action of chitin synthesis inhibitors in arthropods. The authors in [73] conducted a study on the mutation in the CHS1 gene of *Plutella xylostella*. They found that the CRISPR/Cas9 system can effectively clarify the molecular mechanism of *CHS1*-inhibiting bioactive molecules and provide insights on the chitin biosynthesis mechanism. The authors in [73] identified a mutation in the *PxCHS1* gene that conferred resistance to benzoylureas (BPUs) in *P. xylostella. Cydia pomonella* is one of the major pests of pome fruits. The CRISPR/Cas9 system was utilized to target the odorant receptor *CpomOR1* in *C. pomonella*. The *CpomOR1* gene is highly expressed in the antennae and encodes the codlemone receptor, an odorant receptor. The study involved the injection of sgRNA and Cas9 mRNA into eggs at the pre-embryonic stage. The fecundity and fertility of the offspring resulting from the mating of a female possessing the *CpomOR1* gene with a male of normal genetic makeup were observed by [74]. The researchers found that the eggs produced by the offspring were non-viable. This species is a significant agricultural pest in Asia, with a primary diet consisting of corn crops. The developmental gene’s function was elucidated using the CRISPR/Cas9 system. The *Ago1* gene was knocked out in *O. furnacalis* (*OfAgo1*) through a study. The study involved the injection of a sgRNA/Cas9 mRNA mixture into freshly laid eggs. Disruption of cuticle pigment in the seventh abdominal and third thoracic segments occurred in hatched larvae with a dysfunctional *OfAgo1* gene. This study demonstrates the involvement of the gene in cuticle pigmentation [75].

**Figure 3 plants-12-03961-f003:**
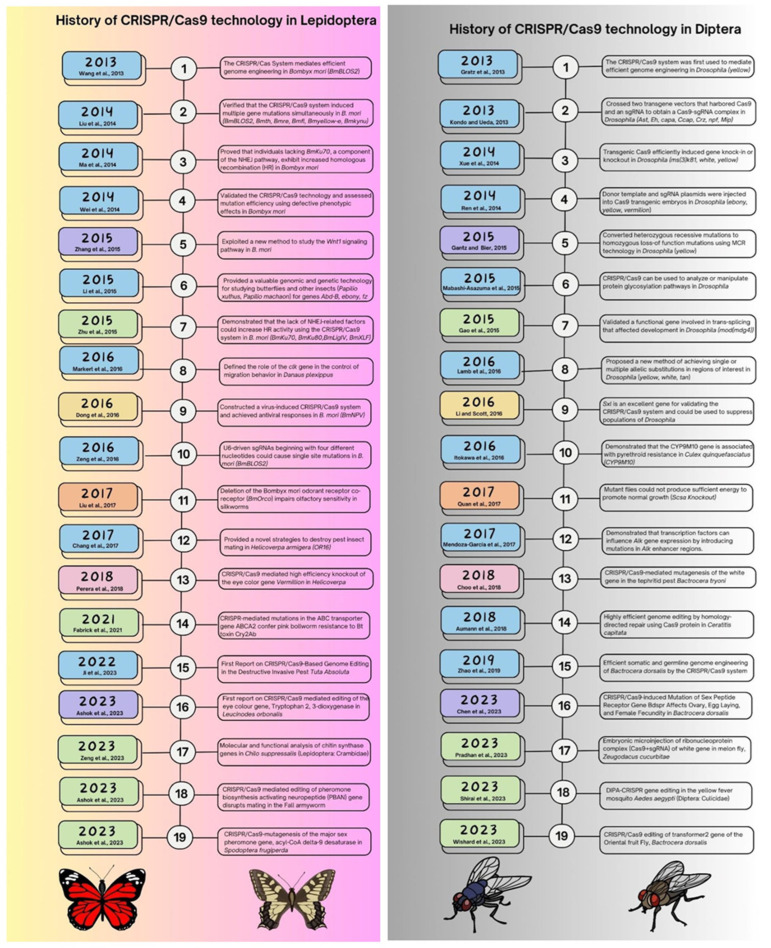
Summarizing the history of CRISPR/Cas9 technology in Lepidopteran and Dipteran insects [63,76,77,78,79,80,81,82,83,84,85,86,87,88,89,90,91,92,93,94,95,96,97,98,99,100,101,102,103,104,105,106,107,108,109,110,111,112].

#### 3.1.2. CRISPR/Cas-Mediated Genome Editing in Dipteran Insects

Diptera is one of the biggest orders of class Insecta comprising true flies having diverse two-winged insects with cosmopolitan distribution across the globe. True flies are the two-winged insects that belong to this order. Because dipterans are widely distributed around the planet, the order is diverse. Their maggots are significant agricultural pests that attack crop plant families, including Solanaceae, Cucurbitaceae, and others. *D. melanogaster*, which serves as the model organism, promoted and enhanced insect genome editing technology via the application of CRISPR/Cas9 [113]. The CRISPR/Cas9 technologically driven excision of 4.6 kb of chromosomal DNA from the *Drosophila* genome was initially reported, and two target sgRNAs and a ssODN (single-stranded oligonucleotide donor) template were used to carry out the deletion in the yellow locus [29]. Additionally, the potential uses and advantages of CRISPR/Cas9 in the development of designer flies were highlighted [114]. In 2013, researchers created a method to boost the frequency of homologous recombination (HR) using a reintegration vector [76], compared the effectiveness of TALEN-based and CRISPR/Cas9-mediated HDR, created an easily screenable platform, and developed three different HDR techniques for site-specific mutagenesis. There were two distinct kinds of parent flies created; one had the Cas9 gene under a germline-specific promoter, while the other had sgRNA expressed constitutively. These parents’ union resulted in offspring with a transmitted mutation in their germlines. Like this, Cas9-positive transgenic flies were given an injection of a gRNA-encoding DNA vector, allowing for the knock-in or deletion of multiple gene targets [115,116]. For instance, the gRNA plasmid and donor repair template were introduced into transgenic Cas9 embryos [117], whereas transgenic embryos harboring a sgRNA and Cas9 were infused with a donor template plasmid. Different methods have been explored to induce HDR in flies. The Cas9 gene, sgRNA, and donor repair template were all given to non-transgenic flies in the form of plasmids [118,119]. The history of using CRISPR/Cas9 technology in dipterans is depicted in Figure 3.

The study conducted by [120] involved a comparison of various methods for facilitating HDR using identical donor plasmid and gRNA. The results suggested the use of non-transgenic individuals resulted in lower frequencies of knock-in events when compared to transgenic individuals. The authors in [77] introduced a method known as mutagenic chain reaction (MCR), which has the potential to result in autocatalytic mutations. This technique was employed to convert mutations from heterozygous to homozygous forms. The utilization of the CRISPR/Cas9 methodology facilitated the targeted induction of a mutation at a specific site within the *Dα6* subunit of the nicotinic acetylcholine receptor (nAChR). According to [121], the Spinosad-based insecticide exhibited a greater resistance level against flies with a *Dα6*-null mutation than those with a site-specific mutation. The study also established a direct correlation between spinosad resistance and the site-specific mutation. Research has also directed attention towards the modified physical characteristics resulting from the inactivation of particular genes through the employment of the CRISPR/Cas9 technology, as reported by [122]. In order to establish a clear correlation between the efficacy of the CRISPR/Cas9 system and the concentration of sgRNA, [123] conducted a study on the yellow gene. However, it was observed that the employment of elevated concentrations of sgRNA resulted in a decrease in the survival rate of the adults targeted. Yu et al., 2013 [124] directed their focus towards the yellow gene and exhibited a significant enhancement in the efficacy of editing.

The CRISPR/Cas9 system was utilized by researchers to introduce mutations at specific sites within the white (*w*) gene of *Drosophila melanogaster* and the sex-lethal (*Sxl*) genes of *Drosophila suzukii*. The observed phenotype resulting from the ocular mutation exhibited reduced efficacy, potentially attributable to the injection of sgRNA and Cas9-encoding plasmid DNA into the flies. Notably, in vivo transcription of plasmid DNA is required in this context, which is not the case with mRNA. The low efficiency observed in this study could potentially be attributed to the specificity of the Drosophila species and the white gene. According to [78], the *Sxl* gene mutant females exhibited anomalous reproductive tissues and genitalia. According to [79], the mutation induced using CRISPR/Cas9 in the alpha subunit of succinyl-CoA synthetase/ligase (*Scsα*) in Drosophila resulted in developmental delays, elevated mortality rates in the absence of food, and compromised locomotor activity, indicating a deficiency in *Scsα*. Therefore, the *Scsα* gene plays a crucial role in the energy metabolism of Drosophila. Asaoka et al., 2016 [125] employed CRISPR/Cas9 methodology to generate *Drosophila melanogaster* specimens that were deficient in linear ubiquitin E3 ligase (*LUBEL*) and reported that the exposure of *LUBEL*-deficient flies to heat resulted in a decrease in climbing ability and a lower rate of survival.

Additionally, the use of the CRISPR/Cas9 system for critical gene mutations in *Drosophila*, such as the clamp gene, troponin C (*TpnC*), *Alk* gene, Sex-lethal (*Sxl*), and white (*w*) gene, has created a solid framework for the creation of long-term pest control methods [80]. The sterile insect technique (SIT), a pest control strategy, calls for raising, sterilizing, and releasing the male insects of the target species. The use of sterile insects for pest management is effective and beneficial to the environment. It is a tool that may be utilized for the basic study of the biology of pests’ reproduction. According to [126], combining SIT with CRISPR-Cas genome editing may result in sterile male strains that are ideal. The white gene’s site-specific editing was made simpler through directly administering the purified Cas9 protein in the *D. suzukii* embryo. A recombinant Cas9 protein could be the best option for producing heritable genomic changes. The authors in [81] created genetically sexing strains of *Bactrocera tryoni* (Froggatt) using a CRISPR/Cas-mediated cascade of frameshift mutations. This study’s findings may be used for SIT-based pest management [81]. In relation to SIT, the sperm marking system aids in the surveillance of the pest population. Sim et al., 2019 [127] created a sperm-marking transgenic strain of *D. suzukii* by utilizing endogenous *D. suzukii* promoters. Cas9 is expressed via the Ds *hsp70* promoter, whereas gRNA is expressed using the U6 promoter of the small nuclear RNA gene. According to the results of this investigation, premade RNPs employed in HRD-based GE were less successful than co-injections of the helper plasmid [128]. In *D. suzukii*, a white (*w-*) gene mutation caused a failure in copulation. Additionally, it resulted in a pigmentation deficit in the testis sheath, which may have been a contributing factor in the copulation failure.

The Tephritid group of Dipterans is recognized as a harmful pest that poses a significant risk to various agricultural crops. The utilization of genetic techniques in pest management exhibits great potential as a means of regulating tephritid populations. This process entails genetic modifications that facilitate the transmission of lethal traits or induce sterility in the insect. The following examples of genetic pest management in Tephritidae are delineated. *Bactrocera dorsalis*, a pernicious insect, is prevalent in various Asian nations. The CRISPR/Cas9 system was utilized to induce mutations in the white and transformer genes. The embryos of *Bactrocera dorsalis* were subjected to co-injection of Cas9 mRNA and sgRNA directed towards the transformer gene. The mutations observed in the transformer gene resulted in a skewed sex ratio towards males, accompanied by the development of atypical reproductive organs both externally and internally. The heritability of the mutations in the subsequent generation indicates that the gene in question may serve as a viable target for the management of this pest, as per the findings of [82]. *Ceratitis capitata*, commonly known as the Mediterranean fruit fly, is a highly polyphagous pest that causes significant economic damage. The utilization of short and single-stranded DNA repair templates in the CRISPR-Cas HDR genome editing technique was highly effective in *C. capitata*, as demonstrated by the eGFP-to-BFP conversion approach. The potential efficiency of targeting a gene that does not exhibit any phenotypic alterations during mutagenic screening, as suggested by [83], may result in significant savings of both time and resources. The authors in [129] utilized the CRISPR-Cas9 system to modify the segmentation paired gene (*Ccprd*) and the eye pigmentation gene white eye (*we*) in *C. capitate* [129].

#### 3.1.3. CRISPR/Cas-Mediated Genome Editing in Hemipteran Insects

The three primary suborders of Hemipteran insects that feed nearly exclusively on plant sap are Auchenorrhyncha (Spittlebugs, Cicadas, Planthoppers, and Leafhoppers), Heteroptera, and Sternorrhyncha (Whiteflies and Aphids) [130]. These insects can survive on a variety of diets. The sucking and piercing mouthparts of hemipteran insects are their distinguishing characteristics. The hemipteran insects have been managed using a variety of management methods. In the realm of pest control and functional investigations of the hemipteran pests, CRISPR/Cas9-mediated genetic modifications have shown amazing promise for offering a straightforward and heritable strategy (Figure 4) [131]. The Brown Planthopper, commonly known as *Nilaparvata lugens*, is a very damaging insect pest that causes significant losses by draining rice plants of their sap and spreading numerous viruses [132]. The cinnabar gene (*Nl-cn*) and the white gene (*Nl-w*) of *N. lugens* were targeted using the CRISPR/Cas9 system in order to overcome limitations in the area of functional genomic investigations. An RNAi-based knockdown study was used to confirm these genes further after they had been deleted. This capacity to introduce precise genetic modifications offers an alternate method for figuring out how genes work and developing fresh pest management strategies for this non-model insect [133].

The Asian citrus psyllid, *Diaphorina citri*, serves as a vector for the pathogenic bacterium *Candidatus Liberibacter* asiaticus (CLas), as reported by [134]. According to [135], the act of feeding by the psyllid on citrus plants results in the transmission of Huanglongbing (HLB) disease. The authors in [136] endeavored to enhance the administration of CRISPR components through the incorporation of Branched Amphiphilic Peptide Capsules (BAPC). The introduction of gene-editing components in close proximity to the insect ovaries resulted in the development of a heritable germline that carried the edited genome in subsequent generations. The study conducted by [136] has successfully circumvented the necessity of microinjecting the eggs. The BAPC-assisted-CRISPR-Cas9 technique was employed by [137] to perform gene editing and gene targeting in adult insects and insect nymphs, specifically in Psyllids (*Diaphorina citri*), Leafhoppers (*Homalodisca vitripennis*), and Whitefly (*Bemisia argentifolii*). The Vermillion (*Vm*) and Thioredoxin (*TXT*) genes were selected for knock-out in the study. The knockouts underwent corresponding alterations in their ocular pigmentation and physiological makeup. The chosen insects in this investigation pose a significant risk to global food security due to their ability to transfer pathogenic microorganisms, such as bacteria and viruses, to plants. The underlying concept driving the approach was to transform vectors into non-vectors. The outcomes were a decreased lifespan, delayed development, diminished reproductive capacity, and altered ocular phenotype. The utilization of BAPC-assisted CRISPR delivery has revolutionized the strategies employed to safeguard food crops against various pathogens and insect vectors, as reported by [137]. In 2020, a study was conducted on *B. tabaci*, wherein a gene editing technique based on CRISPR-Cas9 was developed [138]. The technique involved injecting the CRISPR/Cas9 components into vitellogenic adult females instead of embryos. In the study conducted by [138], the Cas9 protein was conjugated with a peptide ligand called “BtKV” that targets the ovary, resulting in successful and transmissible gene editing in the offspring’s genome.

The gene function of the Neotropical brown stink bug, *Euschistus heros*, was investigated through RNA interference targeting three specific genes, namely yellow (*yel*), tyrosine hydroxylase (*th*), and abnormal wing disc (*awd*). The stink bugs subjected to knockdown of the *awd* gene exhibited wing deformities, while those targeted for the *th* gene displayed a reduction in cuticle pigmentation intensity. The utilization of RNAi techniques for targeting purposes clearly demonstrated that both genes were associated with unique malformed phenotypes, whereas no distinct phenotype was observed for the *yel* gene. Moreover, in order to comprehend the role of this particular gene in the insect, the CRISPR/Cas9-mediated knock-out methodology was devised. Nevertheless, no discernible variations in phenotype were observed between yellow-gene mutant insects and their normal counterparts. In the research conducted, the RNP complex targeting the *yel* gene was introduced into the eggs via microinjection, and subsequent observations of the phenotype studies were carried out on the hatched nymphs of stink bugs [139].

**Figure 4 plants-12-03961-f004:**
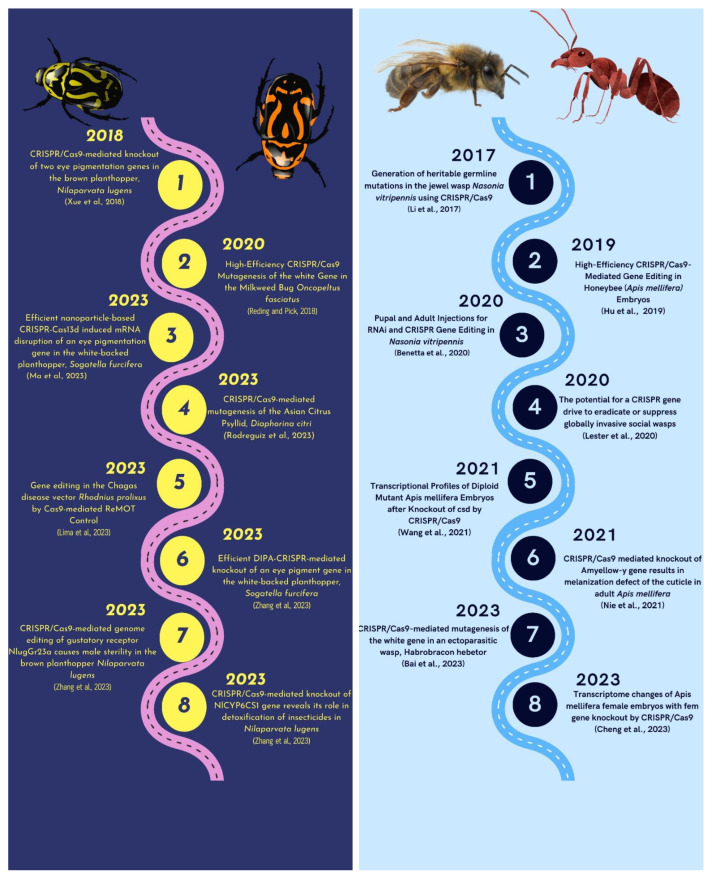
Deciphering the history of the use of CRISPR/Cas9 technology in Hemipteran and Hymenopteran insects [133,140,141,142,143,144,145,146,147,148,149,150,151,152,153,154].

#### 3.1.4. CRISPR/Cas-Mediated Genome Editing in Coleopteran Insects

The Coleoptera is recognized as the largest order among the class Insecta. This sequence presents the prevalent pests that infest stored agricultural commodities. The authors in [9] reported that the forewings of adult coleopteran insects are transformed into rigid elytra. The taxonomic group of beetles encompasses a diverse array of species that occupy a wide range of ecological niches. The Red Flour Beetle, *Tribolium castaneum*, is an agricultural pest that infests stored grains and decreases their nutritional value. The secretions they produce contain carcinogenic compounds such as benzoquinone. The utilization of the genomic database of *T. castaneum*, as presented by [155], has resulted in the advancement of eradication techniques for *T. castaneum* from conventional fumigation to genome editing-based methods. The implementation of novel methodologies has the potential to enhance pest management practices and mitigate the adverse impact on the environment. The utilization of the CRISPR/Cas9 mechanism was implemented in *T. castaneum*, marking its initial application in this particular storage pest [156]. The study conducted by [157] demonstrated that the presence of a mutated *E-cadherin* gene resulted in the occurrence of dorsal closure, and CRISPR-mediated gene targeting and transgene replacement in the beetle were highly effective in achieving a homology-directed knock-in. The GE system can be utilized for multiple species, thereby furnishing fundamental insights for initiating CRISPR-focused transgenic methodologies. According to [158], the potential outcome of this approach is the reduction of developmental gaps between non-model organisms and model organisms.

The utilization of the CRISPR/Cas9 system (Figure 5) for targeted mutagenesis in *Leptinotarsa decemlineata*, commonly known as the Colorado potato beetle (CPB), was first documented by [159]. The vestigial gene (*vest*) was first characterized functionally in CPB through RNA interference. Subsequently, the CRISPR/Cas9 protocol was established in CPB for the purpose of a mutagenesis study. The manifestation of deformed wings as a result of RNA interference was observed once again in the vest gene mutants, and this phenomenon was also replicated through the utilization of CRISPR/Cas9. This study involved the microinjection of the RNP complex for targeted mutagenesis. The targeted mutagenesis through RNAi or CRISPR Cas presented an enhanced pest management approach that is more ecologically sustainable.

#### 3.1.5. CRISPR/Cas-Mediated Genome Editing in Orthopteran Insects

Locusts are significant agricultural pests, and governments have historically implemented policies to eliminate them during early outbreaks. Conventional eradication methods involve the use of synthetic insecticides, resulting in elevated economic and ecological expenses. Locusts are considered significant insects at the molecular level by scientists. A study demonstrated that developmental synchrony is fundamental to gregarious behavior, migration, and copulation. He et al., 2016 [168] discovered that the miRNA gene, *miR-276*, promotes egg development synchronization and regulates locust density. The insect olfactory system is significant, wherein receptors, binding proteins, and degrading enzymes play an important role. Pheromone signals are detected via peripheral tissues and processed in antennal nerve tissues before being integrated into the brain through the olfactory and other sensory organs to regulate insect behaviors, including foraging, feeding, mating, and spawning [169]. The CRISPR/Cas9 technique was employed by researchers (Figure 5) to modify the locust genome in vivo for the purpose of studying its functional genes, wherein an sgRNA was used to target the odorant receptor co-receptor (*Orco*) gene. The microinjection of Cas9-mRNA and *Orco*-sgRNA into locust eggs led to efficient mutation rates in the *Orco* gene. They were able to establish both homozygous and heterozygous mutant lines for *Orco*. [170]. The CRISPR/Cas9 system was employed for genome editing in locusts for the first time. The findings not only proposed novel approaches for locust management but also furnished a framework for utilizing the technology in other orthopteran insects, including crickets.

The human brain performs crucial functions such as learning and memory. Arthropods possess a sophisticated nervous system [171] that facilitates their ability to adjust to diverse external conditions. Insect neurobiology research focuses on the central mechanisms of learning and memory. Numerous studies have demonstrated the associative learning capabilities of insect brains. In mammals, dopamine neurons are believed to have a significant function in facilitating both positive and negative reinforcement. In insects, octopamine and dopamine have been identified as playing a role in appetitive and aversive reinforcement, respectively. Honeybees exhibit proficient olfactory learning and memory capabilities. Vergoz et al., 2007 [172] found that octopamine and dopamine facilitated positive and negative reinforcement, respectively, in olfactory-linked associative learning. In contrast to honeybees, researchers have suggested that dopamine in mushroom bodies plays a role in both positive and negative reinforcement in *Drosophila*, as proposed by [173,174]. Awata et al., 2015 [175] utilized CRISPR/Cas9 technology to knock out the Dop1 gene, responsible for type 1 dopamine receptor, in field cricket, *Gryllus bimaculatus*, in order to address the aforementioned inconsistencies. Awata et al., 2015 [175] found that *Dop1* gene knock-out crickets exhibited impaired aversive learning when subjected to sodium chloride punishment, but their appetitive learning remained intact when presented with water or sucrose reward. The research demonstrated the utility of the CRISPR/Cas9 system in investigating associative learning and gene knock-out in arthropods.

#### 3.1.6. CRISPR/Cas-Mediated Genome Editing in Hymenopteran Insects

According to [176], Hymenoptera is the third most extensive order of the class Insecta, following Coleoptera and Lepidoptera, encompassing various species such as bees, ants, wasps, and sawflies. The hymenopteran order predominantly comprises advantageous pollinators and predatory insects that serve as natural antagonists to pests. *Nasonia vitripennis*, a parasitoid wasp, serves as a crucial biological control agent against pests and is also a highly suitable insect for experimental purposes. Li et al., 2017 [140] employed the CRISPR/Cas9 technique (Figure 4) to target the cinnabar gene responsible for eye pigmentation in *N. vitripennis* specifically. Li et al., 2017 [140] administered sgRNA/Cas9 mixtures into harvested eggs, resulting in the acquisition of efficient and inheritable mutants. The findings of this study indicate that the gene manipulation system is a viable tool for investigating haplo-diploid sex determination, axis pattern formation, and other biological phenomena in *N. vitripennis*.

### 3.2. Precision CRISPR/Cas-Mediated Genome Editing at the Crop Plant Level

The utilization of CRISPR-Cas technologies has the potential to enhance plant vigor, thereby safeguarding and enabling the plants to withstand particular biotic and abiotic stressors, as posited by [177]. The Integrated Pest Management Programme incorporates the maintenance of healthy plants as a crucial component, as insects tend to be attracted to plants that are unhealthy or diseased. The CRISPR-Cas systems can be employed to modify plants in a manner that alters their production of certain enzymes. This modification can serve to either repel insect pests from the plant or attract specific insect predators that can feed on the bug species that are causing harm to the plant [178]. The potential of genome editing in endowing crop plants with insect resistance traits is rapidly increasing. The absence of a well-defined origin of resistance within the genetic makeup of plants has resulted in a reduced focus on modifying crops for the purpose of pest control. The attainment of effective plant regeneration is a crucial determinant for the successful implementation of large-scale CRISPR screens, particularly in crop species characterized by low transformation efficiency, such as maize. In instances of this nature, it is seen that only a limited number of genotypes have the potential to generate significant screening populations. Enhancing the process of plant regeneration not only facilitates the production of larger populations but also expands the range of genetic variation that can be examined. This advancement is crucial for comprehensive CRISPR investigations, enabling researchers to investigate a broader spectrum of genetic variants and phenotypes. The advancement of CRISPR-based agriculture research is expedited by addressing the obstacles related to low transformation efficiency, hence promoting the cultivation of robust and high-yielding plant species. The objective of various initiatives aimed at mitigating this bottleneck is to gather genes from uncharacterized crop plant accessions and their wild counterparts. The lack of comprehension regarding the genetics of resistance characteristics in unclassified accessions has hindered significant progress, as noted by [179]. Conversely, a transgenic approach was employed to incorporate genes conferring insect resistance into crops derived from distant origins, including the Bt genes sourced from bacteria. The introduction of transgenic plant species was met with significant political, ethical, and social resistance due to a dearth of scientific comprehension, as noted by [27]. The primary obstacle in contemporary agriculture pertains to the formulation of an ecologically sustainable breeding approach for crops that can effectively achieve two breeding goals: firstly, the generation of de novo tolerance in the absence of the requisite R-genes, and secondly, the monitoring of pest dynamics through the eradication of insecticide resistance, extermination, or induction of insect sterility. If a suitable food source is present for the offspring of an insect, it will opt to deposit its eggs on the host plant. Plant volatile blends refer to a collection of volatiles that act as signals for insects to identify suitable hosts and locations for laying eggs. Insects employ their remarkably versatile olfactory mechanisms to perceive appropriate host plants by sensing volatile secondary metabolites in plants. Beale et al., 2006 [180] conducted a study that suggests that modifying volatiles via genome editing has the potential to eliminate insects on host plants while simultaneously conferring resistance to the plants. The release of sesquiterpene hydrocarbon I-β-farnesene (Eβf) is triggered in plants when they are infested with aphids. This compound has the ability to reduce the feeding ability of other hosts while simultaneously attracting a parasitic wasp, *Diaeretiella rapae*. The effectiveness of this mechanism has been demonstrated in transgenic plants, where the parasitic wasp has been observed to dominate the aphid population. The manipulation of plant volatile blends through genetic engineering presents a distinct approach to insect control. It is imperative to exercise caution in order to prevent any adverse effects on the populations of beneficial insect species.

It is plausible to augment the host’s resistance to pests through the manipulation of pivotal plant immunity genes, including those that govern the target’s interplay with insect effectors and resistance genes (R-genes). According to [181], while S-genes render plants susceptible to stress, R genes assess the plant’s vulnerability to insect pests and diseases. The manipulation of R and S genes to confer insect resistance in plant species is increasingly recognized as a reliable approach. Insects have been observed to rely on crucial chemical constituents present in plants owing to their growth, immunity, and behavioral patterns, as per the findings of [182] in relation to rice. The utilization of genetic engineering in plants has been exemplified in the context of insect pest resistance through the targeted suppression of the S-genes present in said plants. The deletion of the *CYP71A1* gene responsible for encoding Tryptamine 5-hydroxylase, utilizing the CRISPR-Cas technique, resulted in the conversion of tryptamine to serotonin in plants. This conversion led to a decrease in plant hopper growth. The authors in [182] utilized the CRISPR-Cas9 technology to confer resistance against the striped stem borer (*Chilo suppressalis*) and the brown plant hopper (*Nilaparvata lugens*) on rice. The utilization of the CRISPR-Cas9 technique in *P. tomentosa* Carr. led to the successful creation of endogenous gene mutations in Populus through the concurrent deletion of two endogenous phytoene dehydrogenase (*PDS*) genes, namely *PtoPDS1* and *PtoPDS2*, as reported by [183]. The CRISPR-Cas genome editing techniques have facilitated the augmentation of the endogenous defenses, thereby enabling the amplification of the populus resistance to insects. The beta-1-3 glucanase genes of the golden promise barley cultivar were subjected to CRISPRCas9-mediated genetic modification, resulting in a decrease in the formation of callus in sieve tubes. As per the findings of [27], the growth of *Rhopalosiphum padi*, was negatively impacted due to its inability to access the phloem sap. This also resulted in a disturbance in its preference for specific hosts. Insects possess the ability to identify and selectively prey on specific plants based on their external morphology. Research has indicated that alterations in the pigmentation of plants can impact the selection of a particular host by an insect. The aforementioned statement has been substantiated through the manipulation of the anthocyanin pathway in red-leaf tobacco. The demonstration of gene editing for insect pest tolerance in plants was achieved through the alteration of leaf color. This hindered the insect’s ability to identify the host plant. The excessive presence of anthocyanin pigmentation was responsible for the red hue observed in the leaves. According to [184], the alteration in coloration served as a deterrent for *Helicoverpa armigera* and *Spodoptera litura.* The findings of this research indicate that the utilization of CRISPR-mediated editing for the purpose of pest control, specifically in cases where insects exhibit an inability to identify the host plant, could potentially be addressed through modifications to the anthocyanin pathway. Li et al., 2022 [185] reported that soybean plants harboring the *GmCDPK38* mutant with the *Hap3* deletion exhibited notable resistance against common cutworms. The authors in [186] conducted deletions of the *GmUGT* gene using 1bp and 33bp in soybeans with the aim of enhancing their resistance against *S. litura* and *H. armigera*.

### 3.3. Exploiting Crop Wild Relatives for Insect Resistance via CRISPR-Cas Technology

The incorporation of exogenous genetic material into plant genomes represents a significant regulatory challenge in the realm of transgenic technology, which can be effectively addressed through the application of gene editing techniques. Crop wild relatives (CWRs), which are the ancestors and closely related species of cultivated crops, exhibit resilience to both biotic and abiotic stress factors. However, their yield potential is relatively low. Following the domestication of wild species and selective breeding of plants, the resulting cultivable germplasms and crops exhibited high productivity and versatility in meeting various human demands. However, these crops were found to be vulnerable to insect attacks. The utilization of CRISPR-Cas9 genome editing presents a viable approach to deleting or modifying genes that are responsible for insect susceptibility. Additionally, it is possible to introduce distinctive attributes from CWRs into the cultivated species, thereby generating novel cultivars that exhibit resistance to insects [181]. Two measures can be appropriately taken to implement this. One potential strategy for crop domestication involves the de novo incorporation of insect-resistant traits from wild relatives. Gene-editing methodologies can be employed to modify the targeted agronomic characteristics that are determined using genetic factors. According to [187], there exists empirical evidence to suggest that *Solanum pimpinellifolium*, a type of wild tomato, exhibits resistance towards arthropod insect pests and non-insect pests such as spider mites. According to [188], the application of multiplex CRISPR-Cas editing on six genes in *S. pimpinellifolium* led to the development of a tomato variety that exhibits high yield and resistance to insects and mite pests, all achieved within a single generation. This approach, which relies on the characteristics and molecular mechanisms of a given plant, has the potential to be meticulously implemented in other Crop Wild Relatives (CWRs). The de novo domestication of crop wild relatives (CWRs) has the potential to revolutionize crop improvement through enhancing desirable traits. Subsequently, through the utilization of insect-resistant genes present in Crop Wild Relatives (CWRs), it is possible to modify the genome of domesticated crops. According to [181], it may be possible to conduct an initial investigation of the variability in the sequences of individual insect-sensitive genes across susceptible cultivated germplasms and resilient wild counterparts through the utilization of multiomics techniques. This could be achieved through modifying the genomes of cultivated crops to possess the insect tolerance of wild species. Upon validation against related insects, the resistance genes exhibit potential for successful implementation in gene editing. Li et al., 2016 [189] has indicated that there are opportunities for the emergence of resistance in the genetic makeup of cultivated crops, which are intended to manage insect pests. The utilization of CRISPR-Cas gene-editing-based sequence variation through over-expression or silencing techniques has been proposed as a means to produce insect-resistant phenotypes in commercially valuable crops. Nevertheless, this assertion has yet to be substantiated through empirical evidence.

## 4. Limitations, Future Perspectives and Conclusions

Genome-editing techniques are a type of biotechnological method that involves the targeted modification of a gene using cellular and in vitro mechanisms. The process of genome modification is a natural occurrence that takes place during the course of evolution and is not within our ability to control. The experimental alteration of the genome may have a primary focus on human benefits. This review suggests that the application of crop improvement should be restricted to breeding objectives that are crucial and difficult to attain, given the current heterogeneity. The updated knowledge of the genome of key insects (i5K & other independent projects) and understanding of gene functions in physiological, behavioral, and reproductive science of this diverse group of insects opened up its vast area of application and research. The advancements of gene editing tools and their application in insects to advantageous traits at containment experiments showed greater promise and potentiality. However, public awareness, capacity building, biosafety/off-target effect studies, a proper regulatory framework may provide future field applications and actual benefits of gene silencing and editing in insect pest management. The multiplexing and epigenetic gene editing in insects advances its applications, curtailing costs and multiple pathways for future effective pest management of notorious and invasive insect and mite pests. The potential has been demonstrated in various diverse insects of different groups and orders. To maximize the benefits of this innovation in global agriculture and overcome societal neophobia, it is crucial to take a pragmatic approach that aligns with scientific standards and is endorsed by legislative authorities. The deliberate dissemination of genetic components into wild insect species using CRISPR-Cas technology is a precise and environmentally sustainable approach to pest control. This method involves altering the population’s sex ratio or introducing lethal mutations to reduce the pest population. The possibility of insect resistance arising due to a CRISPR-mediated gene drive is a significant concern, with potential implications at both experimental and theoretical levels, as noted by [190]. According to [191,192], the use of multiplex gene editing has the potential to overcome resistance. Addressing insect resistance issues is a crucial step towards reaching a consensus on the ethical and scientific aspects of this technology. The extensive and/or prolonged cultivation of genetically modified (GE) crops, especially in monocultures, gives rise to concerns over the potential emergence of so-called “better adapted” strains. Genetic modification endeavors to augment favorable characteristics, such as resistance to pests and increased crop productivity. However, the uniformity inherent in monocultures fosters a setting that facilitates the development of specialized adaptations. Over a period of time, the persistent cultivation of genetically engineered (GE) plants has the ability to apply selective pressure on native populations of pests or pathogens; hence, potentially resulting in the emergence of more vigorous and adaptable strains capable of overcoming the engineered resistance mechanisms. The inadvertent outcome has the potential to compromise the long-term viability and efficacy of genetically modified crops, highlighting the importance of diligent oversight and sustainable agricultural methods in order to address and minimize possible hazards linked to the extensive production of genetically engineered organisms.

The use of CRISPR-Cas-edited insects with gene drives has the potential to impact populations and ecosystems significantly. As a result, there are several biosafety concerns associated with introducing these engineered insect pests into the environment. Before being released, a thorough evaluation of potential risks to unintended outcomes is necessary. The authors in [27] suggest that the post-release impacts on beneficial insects can have unforeseen consequences, potentially disrupting food chains and altering community composition. The potential for gene transfers between the target organisms and their non-target relatives can exacerbate the disease. The potential effectiveness of gene-driven technology in eradicating insect pests, insect vectors for viruses, and alien insect species could be realized if the associated risks are adequately managed, particularly with regard to unforeseen environmental consequences. The use of terminator genes and tagged insects for gene flow monitoring is an important measure in ensuring the safe implementation of gene drives and managing associated risks. The use of robotic equipment and artificial intelligence is suggested as another option for managing invasive pests. This approach involves physically eliminating individual pests, as noted by [193]. The effectiveness of robotics may be limited when it comes to dealing with small insects, navigating through uneven terrain, and locating hidden eggs.

The use of CRISPR Cas-based deletion of genes has proven to be a successful method for achieving insect resistance to invasive pests. S gene deletions pose a fundamental problem due to their pleiotropic effects in plants, which contribute to a fitness penalty. The authors in [9] suggest that it may be feasible to maintain plant performance while achieving insect resistance by modifying the binding effector factor instead of the gene itself. The utilization of the CRISPR-Cas technique to confer insect resistance in crop species is expected to be a promising strategy for efficiently introducing genetic traits in cultivated varieties within a shorter timeframe. The statement acknowledges the rapid development of CRISPR-Cas genome-editing technology and its increasing use in agriculture, as noted by [194]. Before implementing insect pest resistance and plant protection through genetic modification, it is essential to have a better comprehension of the gene and genome activities of the species being targeted. The development of Bt technology through recombinant DNA technology has significantly transformed insect management in economic crops such as cotton, maize, soybean, and brinjal. However, with the advent of CRISPR technology, which offers simplicity and multiplexing capabilities, it is possible to replace the current recombinant DNA technology for the insertion of R gene(s) in a faster and more efficient manner. The important examples of CRISPR-Cas mediated Genome Editing for Effective Insect Pest Management are mentioned in Table 1.

The use of CRISPR-Cas technology in the field of entomology has considerable potential, including a wide range of applications in pest management and scientific investigation. Within the field of insect pest management, the use of CRISPR technology allows precise genetic modifications that may effectively bolster resistance mechanisms or regulate population dynamics. The implementation of this innovation has the capacity to significantly transform the field of agriculture, as it aims to reduce the dependence on chemical pesticides while advocating for the use of more environmentally friendly pest management methods. Furthermore, within the realm of insect biology, the use of CRISPR technology enables the exact manipulation of genes, hence providing researchers with the means to untangle the many complications associated with insect behavior, development, and physiology. The influence of technology encompasses disease vector control, whereby there exists the potential to genetically modify mosquitoes to exhibit reduced susceptibility towards harboring and spreading illnesses such as malaria and dengue. Within the field of agriculture, the use of CRISPR technology presents promising opportunities for augmenting the process of pollination and cultivating crops that possess heightened resilience against insect infestations. In the context of the advancing sector, it is essential to provide thorough attention to ethical, ecological, and regulatory aspects in order to guarantee the appropriate and efficient use of CRISPR-based solutions in the area of entomology. The CRISPR Cas genome-editing techniques have had a significant impact on the functional genomics of insects despite being a relatively new technology. The CRISPR-Cas technology enables the manipulation of DNA in various crop and insect species, allowing for the quick and precise alteration, removal, and addition of genetic material. This can potentially confer immunity to insect pests on plants. The technology currently in use for producing crop plants needs improvement to increase their resistance to insect pests. Effective actions need to be taken to safeguard our crops against insect pest infestations and prevent substantial output losses. These measures should be sincere and proactive in nature. The use of CRISPR in crop enhancement projects and the fate of genome-modified products will depend on the decisions made by regulatory authorities around the world. Regulatory systems for novel crop cultivars typically adhere to either product- or process-based regulation. The regulations imposed on CRISPR-based crops will impact both their production costs and the speed at which they are introduced into commercial industries. The statement suggests that regulatory approaches for crops produced through CRISPR-Cas genome editing could be categorized in the same way as those created through classical mutagenesis. This would exempt them from the regulations imposed on genetically modified products. The positive impact of this technology on public perception and its potential for widespread adoption by nations is significant. Several nations have approved CRISPR-edited goods that do not contain transgenes. The potential for a new green revolution in agriculture is anticipated with the use of CRISPR-Cas technology, provided that the adoption of CRISPR products and technological know-how is deregulated in a timely manner and open scientific practice is followed.

## Figures and Tables

**Figure 1 plants-12-03961-f001:**
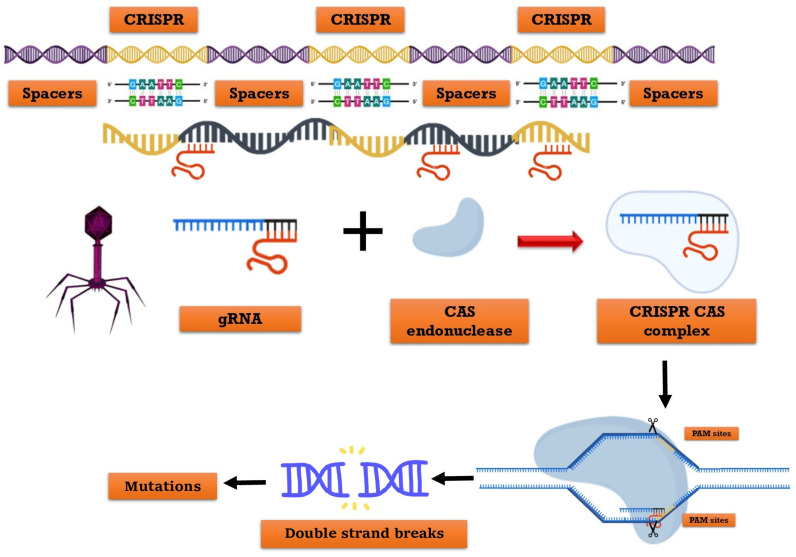
Representing architecture and mechanism of the CRISPR/Cas system. The Constituent elements of the crRNA transcribed from the whole CRISPR array, and tracrRNA transcribed from repeat sequences of the CRISPR array. The gRNA consists of crRNA and tracrRNA. The Cas9 enzyme recognizes PAM (NGG) site and cleaves target DNA sequence between the third and fourth bases near the PAM site.

**Figure 2 plants-12-03961-f002:**
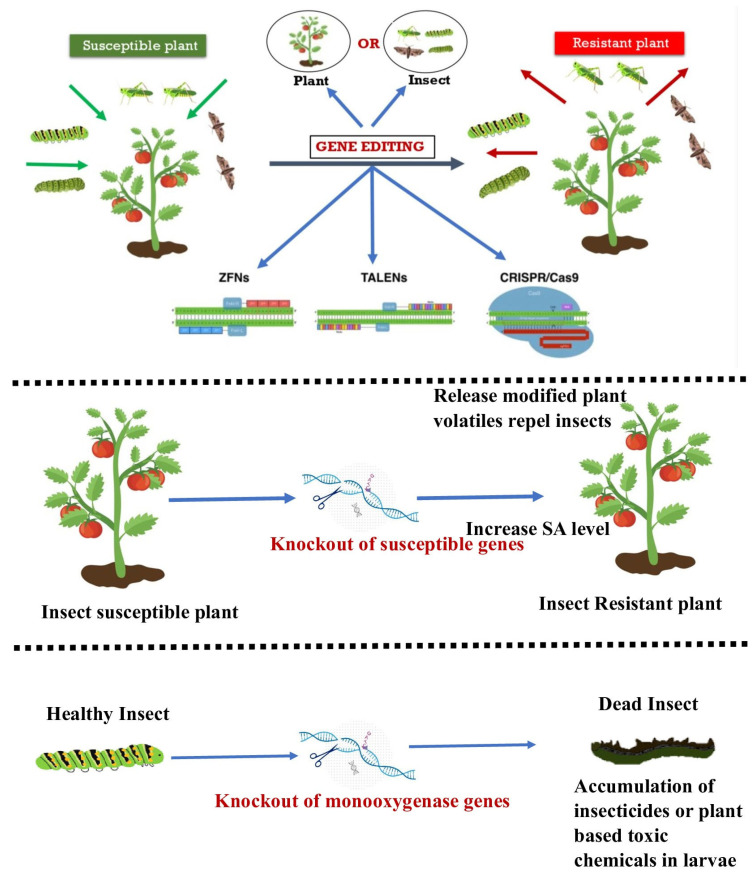
Depicting emerging tools of genome editing for resistance to insect pests. Genome editing of either plant or insect can make the insect susceptible plant to a resistant plant. The potential tools can be ZFNs, TALENs, or CRISPR/Cas9.

**Figure 5 plants-12-03961-f005:**
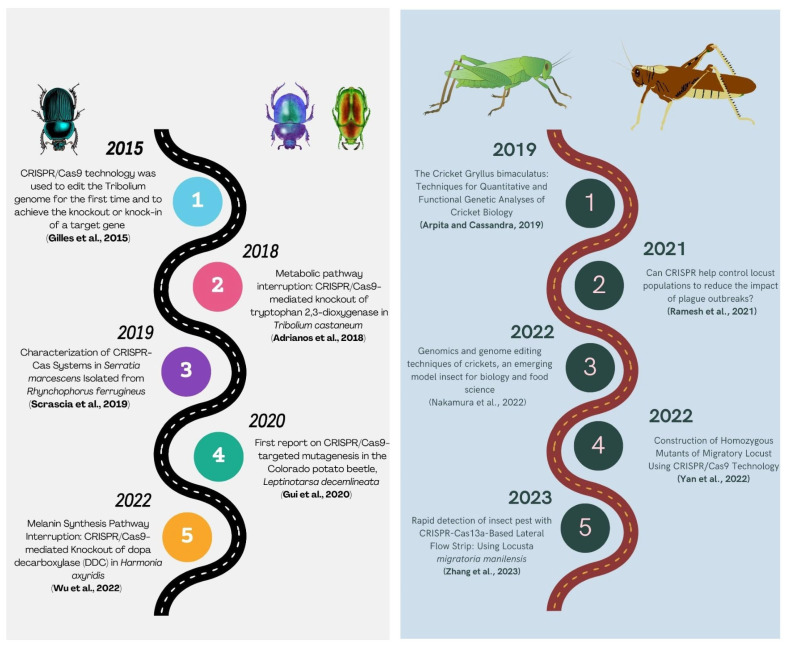
Summarizing the history of CRISPR/Cas9 technology in Coleopteran and Orthopteran insects [157,159,160,161,162,163,164,165,166,167].

**Table 1 plants-12-03961-t001:** CRISPR-Cas mediated Genome Editing for Effective Insect Pest Management.

SL. No.	Name of the Insect	Targeted Gene	Resultant Effect	Delivery Method	References
1	*Drosophila suzukii*	white (*w*) gene	Disruption of the white gene leads to pigmentation deficiency and copulation failure.	Micro-injection	[195,196]
2	*Anastrepha ludens*	*As-transformer-2* (*Astra-2*)	Knocking out of this gene led to sterility in some males and intersexual phenotypes in XX-chromosome females. Thus, exploring this sex-determining gene (Astra-2) can be useful in pest control management.	Micro-injection	[197,198]
3	*Helicoverpa armigera*	*NPC1b*-dietary cholesterol uptake	Limitation in the dietary uptake of cholesterol inhibits the weight gain and food ingestion of the insect.	RNP complex	[61]
4	*Drosophila melanogaster*	*Alk*	Revealed that transcription factors can affect Alk gene expression by establishing mutations in Alk enhancer regions.	Plasmid	[80]
5	*Bactrocera dorsalis*	*White* and *transformer*	CRISPR/Cas9 mediated mutation of white and transformer genes caused various phenotypic effects.	RNP complex	[82]
6	*Plutella xylostella*	*LW*	The results showed weaker phototaxis and reduced locomotion, thus making it a helpful method for pest control.	RNP complex	[199,200]
7	*Spodoptera frugiperda*	*BLOS2, E93,TO*	The developed mutants were helpful in understanding the crucial pathways of *S. frugiperda*, and the strategy can also applied to other invasive pests.	Cas9 protein and multiple sgRNAs	[201,202]
8	*Spodoptera litura*	*SlitBLOS*	The study demonstrated that SlitBLOS2 has a role in the coloration of the integuments, and thus, it provided a marker gene for functional studies and pest control strategies.	Cas9 mRNA and sgRNA	[68]
9	*Bemisia tabaci*	*White*	he method has significantly expanded the capability of CRISPR techniques for whitefly research.	SgRNA + Cas9 protein fused with ovary targeting peptide ligand (BtKV)	[138]
10	*Euschistus heros*	abnormal wing disc (*awd*), tyrosine hydroxylase (*th*), and yellow (*yel*)	Use of RNAi and CRISPR/Cas9 techniques for managing insect pests.	dsRNA, RNP complex	[139]
11	*Locusta migratoria*	*Orco*	Functional genetic studies of locusts by generation of loss-of-function mutation for managing insect pests.	mRNA	[78]
12	*Diaphorina citri*	*ACP-TRX-2*	The method incorporated BAPC-assisted delivery of CRISPR/Cas9 into nymphs and adults, thus resulting in an innovative breakthrough in gene editing and has shown a significant improvement over efforts using injection of eggs.	BAPC-assisted delivery of CRISPR components	[136]
13	*Nilaparvata Lugens*	*Nl-cn* and *Nl-w*	Two genes for eye pigmentation were targeted using CRISPR/Cas9, and the results paved the path for gene-function interrogation.	Cas9 mRNA and sgRNA	[133]
14	*Hyphantria cunea*	*Hcdsx*	Knocked-out Hcdsx gene using CRISPR/Cas9 caused sex-specific sterility, thus making it a pest control method.	sgRNA and Cas9 mRNA	[203,204]
15	*Agrotis ipsilon*	*AiTH*	The AiTH gene knockout using CRISPR/Cas9 caused narrowing in the egg shell.	sgRNA and Cas9 mRNA	[205]
16	*Ostrinia furnacalis*	*OfAgo1*	Mutation in the OfAgo1 gene through CRISPR/Cas9 technology caused cuticle disruption	sgRNA and Cas9 mRNA	[75]

## Data Availability

Data sharing is not applicable to this article as no datasets were generated or analyzed during the review.

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
