# Peer review of "Unveiling the Genetic Symphony: Harnessing CRISPR-Cas Genome Editing for Effective Insect Pest Management"

_plants, 2023, doi:10.3390/plants12233961_

Round 1
Reviewer 1 Report
Comments and Suggestions for Authors
The inclusion of 1/2 more figure/table/scheme would benefit the review. Also to give real-life applications for the technology.
Comments on the Quality of English LanguageThe english is very good, requiring only minor editing
Reviewer 2 Report
Comments and Suggestions for Authors
Plant genome editing (GE) can be generally perceived as a promising tool for a broad range of scientific or commercial purposes, especially in pathogen and pest tolerance. Since its discovery, CRISPR-Cas GE has been applied to enhance disease resistance and abiotic stress tolerance in model plants and in some crops. Despite the great potential and some outstanding results recently mentioned by Li Y. et al. (2022, Front. genome Ed. 4, 987817), the success of plant GE is far from being achieved in practice. But authors of the review made an interesting attempt to present and comment on all current applications of CRISPR-Cas technology to produce crops resistant to the large variety of insect species causing annual loss in agricultural production all over the world.
The review is well constructed. The title, abstract, and keywords clearly reflect the paper's content. The introduction presents the problem clearly and concisely. Later, the paper presents the mechanism of CRISPR-Cas “molecular scissors”, and then discuss the potential use of GE at the level of insect (according to the systematics range of Lepidoptera, Diptera, Hemiptera, Coleoptera, Orthoptera, and Hymenoptera), and later on the level of the plant (comprising main crops).
Generally, to induce disease resistance via GE at the plant level, identification and functional annotation of plant susceptibility (S) and/or resistance (R) genes are essential. This can be only achieved if there is access to whole genome sequences. However, limited or no data is available on the molecular function of S/R genes in many non-model plants (Rato et al. 2021, Transgenic Res. 30, 427–459). Very often, Genes having a structural or a regulatory role as well as cis-regulatory sequences can be edited by CRISPR-Cas, but the possible occurrence of detrimental pleiotropic effects after such manipulations is an additional challenge to consider. In addition, modification of R/T genes requires the development of alternative, and often more complex, approaches based on genome editing/replacement by homologous recombination, base editing, or prime editing.
The authors correctly state the positive and negative aspects of the GE technique applied in order to induce pest resistance mechanisms in crops. In lines 35-38, it has been mentioned that “the article critically analyses and discusses the potential and challenges associated with exploring and utilizing CRISPR-Cas technology for reducing insect pest pressure in crop plants”. However, some missing points of potential threat can be added:
- the widespread and /or long-term planting of GE plants, especially in monocultures, might lead to the appearance of so-called “better adapted” pests,
- when performing extensive CRISPR screens in plants it is crucial to produce a large number of regenerated plants, which was not discussed by the Authors. When studying crops with low transformation efficiency (e.g. maize) only a few genotypes will have the capacity to generate large screening populations. Improved efficiency of plant regeneration will make it possible to scale up CRISPR studies.
The references are complete and the length is commensurate with the paper's content illustrated by five informative figures.
Author Response
REVIEWER#2
Plant genome editing (GE) can be generally perceived as a promising tool for a broad range of scientific or commercial purposes, especially in pathogen and pest tolerance. Since its discovery, CRISPR-Cas GE has been applied to enhance disease resistance and abiotic stress tolerance in model plants and in some crops. Despite the great potential and some outstanding results recently mentioned by Li Y. et al. (2022, Front. genome Ed. 4, 987817), the success of plant GE is far from being achieved in practice. But authors of the review made an interesting attempt to present and comment on all current applications of CRISPR-Cas technology to produce crops resistant to the large variety of insect species causing annual loss in agricultural production all over the world.
Respected Reviewer, thank you for comments, appreciation and mentioning the key points of manuscript.
The review is well constructed. The title, abstract, and keywords clearly reflect the paper's content. The introduction presents the problem clearly and concisely. Later, the paper presents the mechanism of CRISPR-Cas “molecular scissors”, and then discuss the potential use of GE at the level of insect (according to the systematics range of Lepidoptera, Diptera, Hemiptera, Coleoptera, Orthoptera, and Hymenoptera), and later on the level of the plant (comprising main crops).
Respected Reviewer, thank you very much for your valuable comments, appreciation and mentioning the key points of manuscript.
Generally, to induce disease resistance via GE at the plant level, identification and functional annotation of plant susceptibility (S) and/or resistance (R) genes are essential. This can be only achieved if there is access to whole genome sequences. However, limited or no data is available on the molecular function of S/R genes in many non-model plants (Rato et al. 2021, Transgenic Res. 30, 427–459). Very often, Genes having a structural or a regulatory role as well as cis-regulatory sequences can be edited by CRISPR-Cas, but the possible occurrence of detrimental pleiotropic effects after such manipulations is an additional challenge to consider. In addition, modification of R/T genes requires the development of alternative, and often more complex, approaches based on genome editing/replacement by homologous recombination, base editing, or prime editing.
Respected Reviewer, thank you very much for mentioning the key points of manuscript and discussing the important concepts, the suggestions given are incorporated in the revised version of the manuscript.
The authors correctly state the positive and negative aspects of the GE technique applied in order to induce pest resistance mechanisms in crops. In lines 35-38, it has been mentioned that “the article critically analyses and discusses the potential and challenges associated with exploring and utilizing CRISPR-Cas technology for reducing insect pest pressure in crop plants”. However, some missing points of potential threat can be added:
- the widespread and /or long-term planting of GE plants, especially in monocultures, might lead to the appearance of so-called “better adapted” pests,
Respected reviewer the suggested points are discussed in the authors response as well as in the revised version of manuscript.
The extensive and/or prolonged cultivation of genetically modified (GE) crops, especially in monocultures, gives rise to concerns over the potential emergence of so-called "better adapted" strains. Genetic modification endeavors to augment favorable characteristics, such as resistance to pests and increased crop productivity. However, the uniformity inherent in monocultures fosters a setting that facilitates the development of specialized adaptations. Over a period of time, the persistent cultivation of genetically engineered (GE) plants has the ability to apply selective pressure on native populations of pests or pathogens, hence potentially resulting in the emergence of more vigorous and adaptable strains capable of overcoming the engineered resistance mechanisms. The inadvertent outcome has the potential to compromise the long-term viability and efficacy of genetically modified crops, highlighting the importance of diligent oversight and sustainable agricultural methods in order to address and minimize possible hazards linked to the extensive production of genetically engineered organisms.
- when performing extensive CRISPR screens in plants it is crucial to produce a large number of regenerated plants, which was not discussed by the Authors. When studying crops with low transformation efficiency (e.g. maize) only a few genotypes will have the capacity to generate large screening populations. Improved efficiency of plant regeneration will make it possible to scale up CRISPR studies.
Respected reviewer the suggested comments are discussed in the authors response as well as in the revised version of manuscript.
The attainment of effective plant regeneration is a crucial determinant for the successful implementation of large-scale CRISPR screens, particularly in crop species characterized by low transformation efficiency, such as maize. In instances of this nature, it is seen that only a limited number of genotypes have the potential to generate significant screening populations. Enhancing the process of plant regeneration not only facilitates the production of larger populations, but also expands the range of genetic variation that can be examined. This advancement is crucial for comprehensive CRISPR investigations, enabling researchers to investigate a broader spectrum of genetic variants and phenotypes. The advancement of CRISPR-based agriculture research is expedited by addressing the obstacles related to low transformation efficiency, hence promoting the cultivation of robust and high-yielding plant species.
The references are complete and the length is commensurate with the paper's content illustrated by five informative figures.
Respected Reviewer, thank you for comments, appreciation.
Reviewer 3 Report
Comments and Suggestions for Authors
Dear Authors,
I read carefully your submitted article Plants-2642063 , titled "Unveiling the genetic symphony: harnessing CRISPR-Cas genome editing for effective Insect pest Management" by Komal J. et al. It is a very interesting critical review on the CRISPR-cas gene technique and its applications on different groups of Insects. The text is well presented and written in understandable English language. I appreciated your effort to make understanble the concepts , methods and results obtained also to a not specialized audience.
However, I saw some minor errors regarding the citing of references in the text: there are recorded only the numbers belonging to each ones in the final list but the names of the first Authors have been omitted. I suggest to put the first Author's names or both names in case of two Authors. So, the presentation is well improved.I suggest to record all the text's references in this way. Moreover, there is a short sentence unclear (from line223 to 224). See , please, the attached word file with my few notes.
Sincerely

Author Response
REVIEWER#3
Dear Authors,
I read carefully your submitted article Plants-2642063, titled "Unveiling the genetic symphony: harnessing CRISPR-Cas genome editing for effective Insect pest Management" by Komal J. et al. It is a very interesting critical review on the CRISPR-cas gene technique and its applications on different groups of Insects. The text is well presented and written in understandable English language. I appreciated your effort to make understanble the concepts, methods and results obtained also to a not specialized audience.
Respected Reviewer thank you so much for appreciations and comments
However, I saw some minor errors regarding the citing of references in the text: there are recorded only the numbers belonging to each ones in the final list but the names of the first Authors have been omitted. I suggest to put the first Author's names or both names in case of two Authors. So, the presentation is well improved.I suggest to record all the text's references in this way. Moreover, there is a short sentence unclear (from line223 to 224). See, please, the attached word file with my few notes.
Respected Reviewer the references are well organized and arranged according to plants- journal format of MDPI. All the suggestions are incorporated in the current version of the manuscript.
Reviewer 4 Report
Comments and Suggestions for Authors
Gene editing techniques, particularly those associated with CRISPR-CAS, have been a topic of interest for some time, with experimental applications explored in the context of pest management. In this paper, the authors conducted a comprehensive literature review, examining the advancements in using gene editing for pest control, elucidating the challenges faced, and highlighting future directions. The manuscript provides valuable insights, prompting readers to delve deeper into this burgeoning field. Hence, I am inclined to consider it for publication, contingent upon the authors addressing the following concerns.
1. Figure 2 should offer a more detailed depiction of how gene editing can be practically applied in pest management. Although the authors discussed these concepts, the figure lacks sufficient clarity in illustrating them.
2. I believe Figures 3 and 4 could be better organized. The current format lacks essential points and fails to highlight significant information. Frankly, I found the current figures lacking in valuable insights. I recommend that the authors illustrate historical milestones and underscore key events. Moreover, these events should be logically categorized, moving beyond a simple display of literary references.
